# Characterization and Expression of TGF-β Proteins and Receptor in Sea Cucumber (*Holothuria scabra*): Insights into Potential Applications via Molecular Docking Predictions

**DOI:** 10.3390/ijms26146998

**Published:** 2025-07-21

**Authors:** Siriporn Nonkhwao, Jarupa Charoenrit, Chanachon Ratanamungklanon, Lanlalin Sojikul, Supawadee Duangprom, Sineenart Songkoomkrong, Jirawat Saetan, Nipawan Nuemket, Prateep Amonruttanapun, Prasert Sobhon, Napamanee Kornthong

**Affiliations:** 1Chulabhorn International College of Medicine, Rangsit Campus, Thammasat University, Pathumthani 12120, Bangkok, Thailand; siriphorn.nonkhaow@gmail.com (S.N.); jaa.charupa@gmail.com (J.C.); felixavier507@gmail.com (C.R.); lanlalin.soj@dome.tu.ac.th (L.S.); su.duangprom@gmail.com (S.D.); sineenartsong@gmail.com (S.S.); nipawannuemket@gmail.com (N.N.); wanderer_sci@yahoo.com (P.A.); 2Division of Health and Applied Sciences, Faculty of Science, Prince of Songkla University, Hat Yai 90110, Songkhla, Thailand; jisaetan@gmail.com; 3Department of Anatomy, Faculty of Science, Mahidol University, Bangkok 10110, Bangkok, Thailand; prasert.sob@mahidol.ac.th

**Keywords:** *Holothuria scabra*, transforming growth factor-β (TGF-β) superfamily proteins, TGF-β receptor type I, gene expression, molecular docking

## Abstract

*Holothuria scabra* has long been acknowledged in traditional medicine for its therapeutic properties. The transforming growth factor-beta (TGF-β) superfamily is crucial in regulating cellular processes, including differentiation, proliferation, and immune responses. This study marks the first exploration of the gene expression localization, sequence conservation, and functional roles of *H. scabra* TGF-β proteins, specifically activin (*Holsc*Activin), inhibin (*Holsc*Inhibin), and the TGF-β receptor (*Holsc*TGFBR), across various organs. In situ hybridization indicated that *Holsc*Activin and *Holsc*Inhibin are expressed in the intestine, respiratory tree, ovary, testis, and inner body wall. This suggests their roles in nutrient absorption, gas exchange, reproduction, and extracellular matrix remodeling. Notably, *Holsc*TGFBR demonstrated a similar tissue-specific expression pattern, except for its absence in the respiratory tree. Bioinformatics analysis reveals that *Holsc*TGFBR shares significant sequence similarity with *Homsa*TGFBR, especially in regions essential for signal transduction and inhibition. Molecular docking results indicate that *Holsc*Activin may promote receptor activation, while *Holsc*Inhibin functions as a natural antagonist, reflecting the signaling mechanisms of human TGF-β proteins. Interestingly, cross-species ternary complex docking with human TGF-β receptors further supports these findings, showing that *Holsc*Activin moderately engages the receptors, whereas *Holsc*Inhibin exhibits strong binding, suggestive of competitive inhibition. These results indicate that *H. scabra* TGF-β proteins retain the structural and functional features of vertebrate TGF-β ligands, supporting their potential applications as natural modulators in therapeutic and functional food development.

## 1. Introduction

In recent years, research into marine organisms has revealed a wealth of bioactive compounds with significant implications for human health. The sea cucumber (*Holothuria scabra*), often referred to as “the Ginseng of the sea” in Japan, China, Korea, and Hong Kong, is consumed for its reputed medical properties [1]. Bioactive compounds extracted from *H. scabra* have been utilized in traditional medicine and have shown potential applications in wound healing, immune modulation, anti-cancer treatments, protection against degenerative processes, and neuroprotection [2,3,4]. Notably, a previous study identified a conserved family of growth factor and receptor genes in *H. scabra*, which are critical in directing tissue repair and regeneration [5,6]. Although activin and inhibin from *H. scabra* have been reported and characterized, details their expression and proposed function remain limited [6].

The transforming growth factor-β (TGF-β) superfamily, which includes activin and inhibin, plays fundamental roles in cellular signaling, embryonic development, immune regulation, reproduction, extracellular matrix remodeling, and tissue repair in both vertebrates and many invertebrates [7,8]. Activin and inhibin are dimeric glycoproteins within this superfamily of cytokines [9]. Activin typically comprises two β subunits (βA or βB), while inhibin consists of an α subunit linked to a β subunit (βA, βB, or βC) [10,11]. These structural differences result in distinct biological functions. Activin mainly regulates cell proliferation, differentiation, and apoptosis, whereas inhibin functions as a negative regulator of follicle-stimulating hormone (FSH) secretion, playing a crucial role in reproductive feedback regulation [12,13]. Beyond their reproductive roles, activin and inhibin are involved in diverse physiological processes. Activin A, for example, has been shown to suppress epithelial growth in developing gastrointestinal organs in rats and induce apoptosis in rapidly proliferating cells of the forestomach and glandular stomach [14]. In the nervous system, activin is critical for neurogenesis, synaptic plasticity, and neuronal survival [15]. Moreover, activin suppresses the production of nitric oxide (NO), tumor necrosis factor-alpha (TNF-α), and interleukin-6 (IL-6), thereby reducing neuroinflammation and promoting wound healing [16,17]. These findings underscore the broad physiological roles of activin and inhibin, highlighting their potential therapeutic applications in regenerative medicine, metabolic disorders, and neuroprotection [9,17].

TGF-β receptor type I (TGFBR1), also known as activin receptor-like kinase 5 (ALK5), is a serine/threonine kinase vital for mediating TGF-β signaling. Upon ligand binding, TGF-β ligands engage the type II receptor (TGFBR2), which recruits and phosphorylates TGFBR1. This activation initiates downstream signaling cascades that regulate gene expression [18]. Under normal physiological conditions, TGFBR1 signaling is crucial for maintaining cellular homeostasis by modulating cell proliferation, apoptosis, immune responses, and extracellular matrix production. It is also essential for preserving tissue integrity, contributing to processes such as wound healing, tissue repair, and immune tolerance [19]. In cancer cells, TGFBR1 exhibits a dual role depending on the tumor stage. During early tumorigenesis, it functions as a tumor suppressor by inhibiting cell growth and promoting apoptosis. However, in advanced stages, the overexpression or constitutive activation of TGFBR1 can facilitate cancer progression by promoting epithelial mesenchymal transition (EMT), invasion, metastasis, and immune evasion [20,21,22]. This duality has been documented in various malignancies, including breast, pancreatic, colorectal, and liver cancers. Given its role in cancer, TGFBR1 is a therapeutic target, particularly in tumors where TGF-β signaling promotes metastasis and immunosuppression [23].

In this study, we investigated the expression patterns of activin, inhibin, and their receptor (TGFBR1) transcripts in various organs of *H. scabra*, including the intestine, respiratory tree, ovary, testis, and the inner body wall, using In situ hybridization. Structural analyses provided additional evidence for the conservation of TGF-β receptors between *H. scabra* and *Homo sapiens*, particularly in regions critical for ligand and receptor interactions. To explore the potential functions of these proteins in humans, we conducted molecular docking analysis to predict their binding affinity with human TGF-β receptors. This approach helped identify possible cross-species functional similarities. Our findings may contribute to a deeper understanding of TGF-β signaling in marine invertebrates and highlight the biomedical significance of bioactive compounds derived from *H. scabra*.

## 2. Results

### 2.1. In Situ Hybridization of H. scabra Activin and Inhibin

In situ hybridization technique was performed to investigate the localization of activin and inhibin mRNA in the intestine, respiratory tree, testis, ovary, and inner body wall of *H. scabra* [24]. The signals were visualized as distinct blue staining, indicating the sites of mRNA expression. In the intestine, blue staining was prominently observed in epithelial lining of the intestine and respiratory tree. These mRNAs were also detected in the coelomic epithelium and connective tissue of intestine. In the reproductive organs, activin and inhibin localization were found in the spermatocytes of testis and oocytes of ovary. Finally, in the inner wall, blue staining was observed within specific cells and connective tissues (Figure 1A,B).

### 2.2. Characterization and Expression of H. scabra TGF-β Receptor (HolscTGFBR)

The full-length nucleotide sequence of *Holsc*TGFBR, comprising 1850 bp, was successfully amplified, validated, and made available in NCBI with accession number PV173756 (Appendix A). It encodes a protein of 381 amino acids with an estimated molecular weight of approximately 43.2 kDa (Figure 2). Furthermore, for the specific localization of the TGF receptors in the target organs, we found that the *Holsc*TGFBR mRNA was exclusively expressed in the intestinal epithelium, spermatocytes of the testis, oocytes of the ovary, and cells in the inner wall of the *H. scabra* (Figure 3A). The expression of *Holsc*TGFBR was analyzed in the selected tissues using RT–PCR, which produced an amplicon of approximately 207 bp (Figure 3C). The results demonstrated that the *Holsc*TGFBR transcript was widely expressed in the intestine, ovary, testis, and inner body wall, suggesting its involvement in diverse physiological processes within these organs. In contrast, the expression of *Holsc*TGFBR was not detected in the respiratory tree, radial nerve cord, and nerve ring (Figure 3C). The specificity of the RT-PCR assay was validated by the absence of amplification in the non-template (negative) control. Additionally, the expression of 16S rRNA was used as a housekeeping gene to ensure the reliability and accuracy of the observed gene expression profiles.

### 2.3. Bioinformatic Analysis: Amino Acid Sequencing Alignment, Phylogenetic Tree, and Structural Analysis of H. scabra TGF-β Receptor

A phylogenetic analysis of *Holsc*TGFBR compared with the TGF-β receptor I proteins from various species provided valuable insights into its evolutionary relationships and divergence. The analysis demonstrated that *Holsc*TGFBR was closely related to the TGF-β receptors of *Apostichopus japonicus* (*Apoja*TGFBR), *Strongylocentrotus purpuratus* (*Strpu*TGFBR), *Acanthaster planci* (*Acapl*TGFBR), and *Patiria miniata* (*Patmi*TGFBR), indicating a conserved functional role of this receptor within the echinoderm phylum. Notably, *Holsc*TGFBR also exhibited significant similarity to the human TGF-β receptor (*Homsa*TGFBR), suggesting a potential evolutionary link that extends across distant taxa (Figure 4). Sequence alignment revealed a 66.4% identity between *Holsc*TGFBR and *Homsa*TGFBR. Several highly conserved regions were identified, including the GS loop (residues 185–194 in *Homsa*TGFBR and 63–72 in *Holsc*TGFBR), the αGS2 region (195–204 in *Homsa*TGFBR and 73–82 in *Holsc*TGFBR), the phosphate-binding loop (211–217 in *Homsa*TGFBR and 88–94 in *Holsc*TGFBR), the catalytic segment (331–351 in *Homsa*TGFBR and 209–229 in *Holsc*TGFBR), and the activation region (354–375 in *Homsa*TGFBR and 232–253 in *Holsc*TGFBR) (Figure 5A). Furthermore, the three-dimensional structure of *Holsc*TGFBR was predicted using the human Type I TGF-β receptor (PDB ID: 1b6c.3.B) as a structural template. A comparative structural analysis confirmed that key functional domains are conserved and exhibit high structural alignment. These findings support the functional significance of *Holsc*TGFBR in mediating TGF-β receptor activation and inhibition (Figure 5B–D) [25,26].

### 2.4. Three-Dimensional Structural Modeling of H. scabra Activin and Inhibin Proteins

Three-dimensional structural models of *H. scabra* activin and inhibin (*Holsc*Activin and *Holsc*Inhibin) were generated using AlphaFold [27]. The results showed the predicted structures with pLDDT confidence scores of 69.6 (*Holsc*Activin) and 70.1 (*Holsc*Inhibin) and the predicted template modeling (pTM) scores of 0.685 and 0.464, respectively. To assess the structural precision and sequence similarity relative to experimentally determined structures, SWISS-MODEL was employed for comparative analyses. The templates with the highest sequence similarity were identified as *Homsa*GDF8 (human pro-myostatin precursor; 2.6 Å resolution; PDB ID: 5ntu.1.A) for *Holsc*Activin and *Homsa*TGFβ (human pro-TGF-β1; PDB ID: 5vqp.1.A) for *Holsc*Inhibin [28,29].

The structural superposition of the monomeric growth factor domains of *Homsa*GDF8 and *Holsc*Activin is illustrated in Figure 6A. In this comparison, *Homsa*GDF8 is depicted with a gray ribbon (pro-domain) and a green ribbon (mature domain), while *Holsc*Activin is illustrated with a pale pink ribbon (pro-domain) and a deep pink ribbon (mature domain). The mature domain of *Holsc*Activin aligns closely with *Homsa*GDF8, preserving the canonical cystine knot fold, critical for receptor binding and signaling. The pro-domain structures exhibit moderate overlap, including subdomains such as the forearm, latency lasso, and arm regions, suggesting conserved latency control architecture despite evolutionary divergence. Sequence analysis indicates an amino acid identity of 30.43% between *Homsa*GDF8 and *Holsc*Activin, reflecting moderate conservation, typical of TGF-β family orthologs across metazoans. Additionally, cysteine residues essential for disulfide bond formation are strictly conserved, ensuring the correct stability of the tertiary structure. The red triangle in Figure 6B indicates the N-terminal start site of the *Holsc*Activin mature domain (S316–F430), showing slight positional variation within the structure. Moreover, mature *Holsc*Activin monomer consists of 114 amino acids and forms homodimers with a molecular mass of approximately 25.7 kDa.

Figure 6C presents the structural superposition of *Homsa*TGFβ (depicted in pale blue for the pro-domain and deep blue for the mature domain) with *Holsc*Inhibin (tan for the pro-domain and orange for the mature domain). The mature domain of *Holsc*Inhibin aligns well with that of *Homsa*TGFβ, maintaining the characteristic β-strand and α-helix arrangements of TGF-β family growth factors [30,31,32]. However, notable structural differences are observed in the pro-domain, particularly within the latency lasso and arm regions, indicating species-specific adaptations that may influence latency mechanisms. Sequence alignment reveals an amino acid identity of 28.32% between *Homsa*TGFβ and *Holsc*Inhibin. The mature domain of *Holsc*Inhibin spans from V258 to Y371, consisting of 114 amino acids, with a predicted molecular weight of approximately 25.3 kDa for homodimers. Furthermore, the cysteine residues crucial for cystine knot motif formation are conserved, ensuring structural integrity (Figure 6D).

### 2.5. Molecular Docking of H. scabra TGF-β Proteins and TGF-β Receptors

To investigate the potential role of *H. scabra* TGF-β ligands (*Holsc*Activin and *Holsc*Inhibin) in signaling through the *H. scabra* type I transmembrane serine/threonine kinase receptor (*Holsc*TGFBR), molecular docking analyses were performed. The complex with the highest coefficient-weighted score, representing the most stable binding interaction, was selected (Table 1). The docking complexes of *Holsc*Activin and *Holsc*Inhibin with *Holsc*TGFBR, using human TGFBR structure (PDB ID: 1b6c.3.B) as a template, are shown in Figure 7A,B. *Holsc*Activin (depicted as a deep pink ribbon) interacts with the extracellular domain of *Holsc*TGFBR (light blue ribbon), forming multiple stabilizing interactions. Hydrogen bonding interactions (yellow dotted lines) are observed between key *Holsc*Activin residues, including H368, G371, S370, R379, F330, T397, C428, and Q378, and receptor residues such as R77, R81, E117, R118, E125, N145, D244, I245, and Q247. The mapping of the interaction interface (middle panel) reveals direct contact residues forming both polar and hydrophobic interactions, underscoring their contribution to binding affinity and complex stabilization. Importantly, *Holsc*Activin did not bind to the key functional regions of *Holsc*TGFBR, including the serine/threonine-rich GS loop (residues 63EGTGSGSGLP72), the αC-helix of the kinase domain involved in structural stabilization, the active site residue E123, and the critical ATP-binding activation segment residue R250, which are essential for receptor activation and signal transduction (Figure 7A) [26,29]. In Figure 7B, the *Holsc*Inhibin–*Holsc*TGFBR complex is shown, where *Holsc*Inhibin (orange ribbon) binds the receptor (light blue ribbon) at an overlapping but slightly shifted site compared to *Holsc*Activin. Hydrogen bonding interactions are identified between *Holsc*Inhibin residues L332, V338, K340, N349, K358, S350, E356, Y371, and receptor residues, including E58, K62, T65, S67, G66, S69, R77, N145, E125, R235, and D244. The critical interaction is that K340 and V338 of *Holsc*Inhibin bind to T65, G66, S67, and S69, which are in the serine/threonine-rich GS loop (Figure 7B). This interaction may physically block the access of the type II receptor kinase (TβRII) and prevent phosphorylation, thereby inhibiting the receptor activation [25,33]. Similarly, in humans, inhibin functions as a natural antagonist that prevents activin from binding to and activating TGF-β receptors, specifically TGFBR1 [26].

### 2.6. Ternary Complex Docking Analysis of Human TGF-β1 and H. scabra TGF-β Ligands with Human TGFβ Receptors

To analyze the sequence conservation and receptor-binding residues among human TGF-β1, *Holsc*Activin, and *Holsc*Inhibin, multiple sequence alignment was performed (Figure 8A). The results reveal that all three proteins retain conserved cysteine residues essential for cystine knot motif formation, which underpins tertiary structure stability and biological function, consistent with the structural requirements of TGF-β superfamily ligands. Residues highlighted in cyan and gray represent amino acids involved in hydrogen bonding with *Homsa*TGFBR2 and *Homsa*TGFBR1, respectively, as identified from structural docking analyses. For *Homsa*TGF-β1A, critical residues forming hydrogen bonds with TGFBR2 include W30, Y91, R94, and K97, consistent with previously reported receptor-binding interfaces [34,35].

The ternary complex of *Homsa*TGF-β1 (red), *Homsa*TGFBR2 (blue), and *Homsa*TGFBR1 (gray) was used as an experimental template. Docking analysis yielded a binding energy of −1054.2 kcal/mol (Table 1). The complex forms multiple hydrogen bonds, including the residues R25, K31, H34, Y91, G93, R94, and R95 of the TGF-β1A interacting with D32, S49, E55, D118, and E119 of TGFBR2. The residues C78 and P80 of TGF-β1A also form hydrogen bonds with D57 and R58 of TGFBR1. Additionally, interactions involved F24, Q26, E59, and D118 of TGFBR2 binding to T18, K19, and D20 of TGFBR1. A central hydrophobic interface was identified, comprising residues I22, L28, W30, W32, A41, F43, V92, and M104 on TGF-β1, which engage hydrophobic patches on TGFBR2 and TGFBR1 (Figure 8B). This hydrophobic core contributes to stable receptor bridging and activation. The *Holsc*Activin ligand (warm pink ribbon) formed a ternary complex with *Homsa*TGFBR2 and *Homsa*TGFBR1, exhibiting a docking energy of −930.5 kcal/mol.

*Holsc*Activin residues W341, Q378, D409, P419, and E420 form hydrogen bonds with F35, A21, N68, D69, E70, and N71 of TGFBR2. Residues R356, S362, E377, R379, G387, and E391 interact with C14, H15, T18, T35, D39, K40, H43, and F60 of TGFBR1. Additionally, interactions include V22, K29, S116, D118, E119, and N123 of TGFBR2 binding to C17, K19, D20, D27, A49, and D52 of TGFBR1. Hydrophobic residues, including F406, W343, M399, F354, F430, F372, I388, L361, and I375, form significant interfacial contacts with TGFBR2 and TGFBR1 (Figure 8C), stabilizing the complex despite weaker binding energy compared to endogenous TGF-β1. The *Holsc*Inhibin ligand (orange ribbon) forms a stable ternary complex with a docking energy of −1002.3 kcal/mol, comparable to endogenous TGF-β1. Hydrogen bonding interactions involve the N330, K340, R351, and K358 residues of *Holsc*Inhibin forming bonds with D32, V33, T51, S52, R66, N68, N71, T73, and D118 of TGFBR2. Moreover, the ligand residues C229, Q300, N314, C335, and A337 bind with T35, D39, and T73 of TGFBR1. The inter-receptor contacts include K23, P25, K97, S117, D118, and E119 of TGFBR2 binding to K40, I42, S33, N44, D52, and T72 of TGFBR1. Extensive hydrophobic contacts are observed with residues W287, W285, M319, L332, M331, L363, and F348, enhancing complex stability and receptor occupancy (Figure 8D).

**Table 1 ijms-26-06998-t001:** The highest coefficient weight scores from PIPER energy calculations were obtained for the complexes of human and *H. scabra* TGF-β ligands with their respective TGFBR receptors, following docking analysis using ClusPro 2.0.

Complex Model	Cluster Members	Weight Score ^a^
*Holsc*Activin–*Holsc*TGFBR	57	−958.0
*Holsc*Inhibin–*Holsc*TGFBR	70	−887.7
*Homsa*TGF-β1–*Homsa*TGFBR2	107	−804.7
*Holsc*Activin–*Homsa*TGFBR2	45	−748.4
*Holsc*Inhibin–*Homsa*TGFBR2	36	−844.2
*Homsa*TGF-β1–*Homsa*TGFBR2–*Homsa*TGFBR1	81	−1054.2
*Holsc*Activin–*Homsa*TGFBR2–*Homsa*TGFBR1	72	−930.5
*Holsc*Inhibin–*Homsa*TGFBR2–*Homsa*TGFBR1	95	−1002.3

^a^ The PIPER protein interaction energy was calculated from *E* = w_1_*E_rep_* + w_2_*E_attr_* + w_3_*E_elec_* + w_4_*E_DARS_*, where *E_rep_* and *E_attr_* represent the repulsive and attractive components of the van der Waals interaction energy, and *E_elec_* indicates an electrostatic energy term. *E_DARS_* reveals the pairwise structure-based potential constructed using the decoys as the reference state (DARS) method [36].

## 3. Discussion

This study is the first to investigate the localization, sequence conservation, structural alignment, and potential functional roles of *H. scabra* TGF-β proteins (activin and inhibin) across various organs, including the intestine, respiratory tree, ovary, testis, and inner body wall [5,24]. Additionally, we examined the specific gene expression and localization of the *H. scabra* TGF-β receptor. A comparative sequence alignment was also conducted between *H. scabra* TGF-β receptors and those of multiple species, particularly *Homo sapiens*. Furthermore, molecular docking was performed to assess the relevance of *H. scabra* TGF-β proteins in binding adaptively to the human TGF-β receptor. These findings provided fundamental insights into the potential biomedical applications of *H. scabra*-derived proteins [2,29,37,38].

For specific localization investigation, the in situ hybridization results indicated that the *H. scabra* TGF-β genes, including activin and inhibin, were localized across all targeted organs. In the intestine, the mRNA for activin and inhibin were prominently observed in the epithelial layer, aligning with their roles in nutrient absorption and digestion [39]. In the respiratory tree, signals were detected in the lining of the respiratory tubules, suggesting a possible role in gas exchange or osmoregulation [40]. In the testis, intense signals were localized to somatic cells surrounding the germinal epithelium, indicating a regulatory role in spermatogenesis. Similarly, signals were concentrated in the follicular layer surrounding developing oocytes, implying involvement in oocyte maturation [41,42]. Notably, the TGF-β genes were expressed in the inner wall, suggesting their presence in connective tissue and specific cells, which may imply a role in extracellular matrix remodeling, structural integrity, and repair processes [43,44]. These observations are consistent with the known functions of TGF-β family members in cellular communication, matrix control, and tissue homeostasis [45]. Consequently, the findings suggest that activin and inhibin may play a potential role in extracellular matrix remodeling and maintaining structural integrity [46,47,48].

Interestingly, the *H. scabra* TGF-β receptor (*Holsc*TGFBR) expression exhibited high sequence identity with the human TGF-β receptor type I, which is involved in signal transduction [25]. The tissue-specific expression pattern of *Holsc*TGFBR indicated its functional significance in important physiological processes. Its exclusive expression in the intestinal epithelium, spermatocytes, oocytes, and inner body wall implies potential roles in digestion, reproduction, and tissue remodeling, in line with the roles of TGF-β signaling in invertebrate development and homeostasis [49]. The widespread expression of *Holsc*TGFBR in reproductive and digestive tissues was confirmed by RT-PCR. The absence of expression in neural tissues, such as the radial nerve cord and nerve ring, suggests a tissue-specific regulatory role rather than direct involvement in neural signaling. These findings highlight *Holsc*TGFBR as a key regulator in *H. scabra* physiology, warranting further investigation into its downstream signaling pathways and functional implications [50].

The phylogenetic and structural analyses of *Holsc*TGFBR underscore its evolutionary conservation and potential functional significance. Phylogenetic comparisons show a close relationship between *Holsc*TGFBR and TGF-β receptor homologs in other echinoderms. This similarity suggests a conserved role in fundamental biological processes, such as development, regeneration, and immune regulation, consistent with the TGF-β signaling functions in marine invertebrates [51]. Remarkably, *Holsc*TGFBR exhibits significant sequence similarity to the human TGF-β type I receptor (*Homsa*TGFBR), particularly in key functional regions like the GS loop, αGS regions, phosphate-binding loop, catalytic segment, and activation region. This conservation implies that *Holsc*TGFBR likely employs similar phosphorylation-dependent mechanisms for receptor activation as observed in vertebrates. The preserved catalytic segment and phosphate-binding loop further support its potential kinase activity in signal transduction [52].

We investigated the structural conservation and receptor-binding potential of two *H. scabra* TGF-β superfamily ligands, *Holsc*Activin and *Holsc*Inhibin, to assess their functional parallels with vertebrate TGF-β family members. AlphaFold modeling indicated moderate confidence, with both proteins displaying overall structural similarity to canonical TGF-β templates. SWISS-MODEL analyses identified *Homsa*GDF8 (pro-myostatin precursor) as the closest homolog for *Holsc*Activin and *Homsa*TGFβ1 for *Holsc*Inhibin, consistent with expected divergence across metazoan orthologs. Importantly, both *Holsc*Activin and *Holsc*Inhibin retained all cysteine residues necessary for cystine knot motif formation, ensuring correct folding and structural stability [35,53]. Structural superposition showed that mature *Holsc*Activin aligned closely with *Homsa*GDF8, preserving the canonical cysteine knot fold critical for receptor binding. *Holsc*Inhibin similarly maintained the β-strand and α-helix arrangements characteristic of mature *Homsa*TGFβ. While their mature domains were well conserved, notable species-specific differences were observed in pro-domain regions, particularly in *Holsc*Inhibin. These included variations within the forearm, latency lasso, and arm subdomains, which regulate ligand latency and activation mechanisms, suggesting evolutionary adaptations that may influence receptor affinity or signaling dynamics [34,54,55]. These findings indicate that *Holsc*Activin and *Holsc*Inhibin share conserved structural features with vertebrate TGF-β family ligands, supporting their potential roles in *H. scabra* cellular regulation, including growth, differentiation, and immune responses [53,56].

For molecular docking, we investigated the molecular interactions of *Holsc*Activin and *Holsc*Inhibin with their putative receptor (*Holsc*TGFBR) to evaluate their potential signaling and inhibitory properties within the TGF-β superfamily framework. *Holsc*Activin docked onto *Holsc*TGFBR, forming multiple hydrogen bonds with the receptor residues, including R77, E117, and R118, and stabilizing hydrophobic contacts while avoiding critical catalytic regions, activation segments, and the GS loop. This indicated that *Holsc*Activin may function similarly to endogenous activin ligands as well as activating TGF-β signaling [57,58]. Conversely, *Holsc*Inhibin showed binding to receptor regions overlapping with *Holsc*Activin but critically engaged the serine/threonine-rich GS loop through K340 and V338 residues interacting with T65, G66, S67, and S69. Such interactions could physically obstruct type II receptor recruitment and block receptor activation, consistent with the established inhibitory mechanism of vertebrate inhibins, which prevent activin-induced signaling by sequestering receptor sites [38]. A balance between these ligands controls cell proliferation and differentiation and plays an important role in normal development and homeostasis [59,60].

Furthermore, we concentrated on these TGF-β proteins from the inner wall, as they possess the potential to be developed into functional food and medical applications [25,55]. Interestingly, cross-species docking analysis further provided insights into potential functional conservation between *H. scabra* and human TGF-β proteins. Ternary complex docking with human receptors (*Homsa*TGFBR1 and *Homsa*TGFBR2) further elucidated the competitive binding potential of *Holsc*Activin and *Holsc*Inhibin. The crystal structure of the *Homsa*TGF–β1–*Homsa*TGFBR2–*Homsa*TGFBR1 ternary complex (PDB ID: 3KFD) served as a template, enabling the structural modeling to predict their ability to mimic endogenous TGF-β ligand binding and receptor activation [34]. Molecular docking analysis revealed distinct roles of *H. scabra* TGF-β proteins in receptor activation and inhibition. Ternary complex docking with human TGF-β receptors further supported these interpretations. Endogenous human TGF-β1 exhibited the strongest binding energy (−1054.2 kcal/mol) with extensive hydrogen bonding and hydrophobic interactions bridging TGFBR2 and TGFBR1, consistent with its potent activation capability [37]. Importantly, the complete complex exhibited both hydrogen bonding and hydrophobic interactions at regions and specific residues consistent with those in the experimental crystal structure [34]. This approach can therefore be applied to *Holsc*Activin and *Holsc*Inhibin for cross-species docking analyses, providing further insights into the potential function of *H. scabra* proteins. *Holsc*Activin showed moderate binding (−930.5 kcal/mol), forming multiple interfacial interactions yet lacking the extensive hydrophobic core stabilizing endogenous TGF-β1, suggesting partial receptor engagement. In contrast, *Holsc*Inhibin displayed a docking energy (−1002.3 kcal/mol) approaching that of TGF-β1, forming robust inter-receptor contacts and occupying the receptor interface in a manner indicative of competitive inhibition without activation. These results suggested that these ligands can mimic endogenous TGF-β binding modes and potentially modulate receptor-mediated signal transduction [35,37].

In short, we predicted the functional conservation of *H. scabra* TGF-β proteins, with *Holsc*Activin functioning as a potential agonist and *Holsc*Inhibin serving as an antagonist of activin signaling in human TGF-β receptor responses [59,60]. These findings will lead to medical application investigations utilizing *H. scabra* as a naturally functioning resource. Nonetheless, further studies are required to assess their efficacy, safety, and interactions with human receptors and signaling pathways [61]. Additionally, incorporating these proteins into nanoparticle-based or other targeted delivery systems may enhance their therapeutic potential. This research supports the broader pursuit of nature-sourced approaches for novel functional food as well as biomedical applications in the future [62,63].

## 4. Materials and Methods

### 4.1. Ethics Statement

All animals involved in this experiment were approved by the Institutional Animal Care and Use Committee (IACUC) of Thammasat University (Protocol Number 019/2561) according to the established guidelines for scientific research. In this study, we aimed to investigate the distribution of transforming growth factor superfamily (TGF-β), including activin and inhibin, in *H. scabra*. Wild specimens (weighing 300–500 g, with random sexes) were collected from Krabi Province, Thailand, following Songkoomkrong et al., 2024 [4]. Subsequently, tissue samples of *H. scabra*, including the intestine, respiratory tree, gonads, and inner wall, were harvested and immediately frozen in liquid nitrogen or fixed in 4% paraformaldehyde. The frozen samples were kept at −80 °C for later RNA extraction, while the fixed samples were processed for histological examination and in situ hybridization [64].

### 4.2. In Situ Hybridization of the Activin, Inhibin, and TGF-β Receptor Type I in H. scraba

The DIG-labeled sense and anti-sense RNA probes for activin and inhibin were derived from the study by Kornthong et al., 2021 [6,20]. Tissue slides of all organs were routinely prepared using the paraffin method. Following de-paraffinization and rehydration, the slides were immersed in PBS containing DEPC, followed by PBS with 0.3% Tween-20 (PBST), and then incubated with Proteinase K (Roche, Mannheim, Germany) in 1× TNE buffer (500 mM NaCl, 10 mM Tris, 1 mM EDTA, pH 8.0). The slides were washed with 2X Sodium Saline Citrate (SSC) and fixed in 4% paraformaldehyde. For the hybridization procedure, the tissues were incubated in the hybridization solution with RNA probes for *Holsc*Activin, *Holsc*Inhibin, and TGF-β receptor type I (*Holsc*TGFBR) overnight at 52 °C. Afterward, the slides were thoroughly washed with 4×, 2×, and 1× SSC solution and blocked for non-specific binding with maleic acid buffer containing 2% sheep serum at room temperature for 3–4 h. The slides were then incubated with a 1:500 dilution of AP-conjugated anti-DIG antibody overnight at 4 °C. After washing, the tissue slides were stained using the NBT/BCIP detection system. Positive expression sites of activin and inhibin were examined and photographed using a Leica compound microscope (Leica, Wetzlar, Germany). The complete hybridization procedure followed the methodology described by Kornthong et al., 2021 [6].

### 4.3. Bioinformatic Analysis of Activin, Inhibin, and TGF-β Receptor Type I in H. scraba and Other Echinoderms

As a previous report, the transcriptome data set of growth factor transcripts in various organs of *H. scabra* is available in NCBI (Accession no: MW728942 to MW728950) [6,25]. The translational amino acid sequences of the *Holsc*Activin (accession no: MW728947), *Holsc*Inhibin (accession no: MW728945), and *Holsc*TGFBR (accession no: PV173756) were identified using BLASTx (https://blast.ncbi.nlm.nih.gov/Blast.cgi, accessed on 28 April 2024). The molecular mass of mature proteins was calculated using the online software ExPASy Peptide Mass program (http://web.expasy.org/compute_pi/, accessed on 30 April 2024). The analysis of multiple amino acid sequence alignments of *Holsc*TGFBR compared to other species was performed using MEGA11 software, utilizing MUSCLE for alignment. The conserved amino acids were generated with Multiple Align Show (https://www.bioinformatics.org/SMS/multi_align.html, accessed on 3 May 2024). Phylogenetic tree analysis was conducted with MEGA11 software and modified by iTOL v6 software (https://itol.embl.de/upload.cgi, accessed on 3 May 2024).

### 4.4. Tissue Distribution of the TGF-β Receptor Type I mRNA in H. scabra

The collected tissue samples were extracted for total RNA extraction. The concentration and purity of each RNA sample were determined using a nanodrop spectrophotometer at wavelengths of 260 and 280 nm (ND-3800-OD Nano DOT Microspectrophotometer, Hercuvan Lab Systems, Brixton, UK) [65]. cDNA synthesis was then carried out using the QuantiNova Reverse Transcription Kit (Qiagen, Hilden, Germany), and the resulting cDNA served as a template for PCR with gene-specific primers, as shown in Table 2.

### 4.5. Structural Modeling Prediction and Sequence Analysis

Full-length sequences of *Holsc*Activin and *Holsc*Inhibin were obtained from previously published transcriptome data [6]. The three-dimensional structures of both proteins were utilized using AlphaFold2, accessible on https://colab.research.google.com/github/sokrypton/ColabFold/blob/main/AlphaFold2.ipynb, accessed on 4 July 2025 [27]. The predicted local distance difference test (pLDDT) was used to assess model confidence, while the predicted aligned error (PAE) estimated positional uncertainties for each amino acid. Modeling parameters included 20 recycles and a root mean square deviation (RMSD) tolerance of 0.5 Å. For single-chain predictions, ranking was based on pLDDT scores, whereas complex models were ranked by predicted TM-score. Five models were generated per run, and their respective pTM and pLDDT scores were recorded. Additional structural features were retrieved using SWISS-MODEL (https://swissmodel.expasy.org/, accessed on 5 July 2025). After predicting the 3D structures, *Holsc*Activin and *Holsc*Inhibin were analyzed and structurally aligned using the crystal structures of the closest human homologs as templates, including *Homsa*GDF8 (human pro-myostatin precursor; 2.6 Å resolution, PDB ID: 5ntu.1.A) for *Holsc*Activin and *Homsa*TGFβ (human pro-TGF-β1, PDB ID: 5vqp.1.A) for *Holsc*Inhibin. Additionally, the 3D structure of *Holsc*TGFBR was generated by using the SWISS-MODEL, accessed on 15 May 2024, using *Homsa*TGFBR (PDB ID: 1b6c.3.B) as a template. Refinement and annotation were performed using Chimera UCSF software (version 1.6.1).

### 4.6. Molecular Docking

Complex structures of *Holsc*Activin and *Hols*cInhibin with the *Holsc*TGFBR were docked using the cytoplasmic domain of the type I TGF-β receptor (PDB ID: 1b6c) as a complex structural template. For ternary complex docking, *Homsa*TGFβ was initially docked with *Homsa*TGFBR2. The complexes with the most favorable (lowest) energy scores were selected, and *Homsa*TGFBR1 was subsequently recruited to form the complete ternary complex. The crystal structure of the ternary complex (PDB ID: 3KFD) was used as the experimental template. This protocol was similarly applied using *Holsc*Activin and *Holsc*Inhibin in place of *Homsa*TGFβ. All molecular docking complexes were performed on ClusPro 2.0, a widely recognized server for protein–protein docking (https://cluspro.org/help.php, accessed on 6 July 2025). The interaction energy of the protein complexes is based on the PIPER protein interaction energy, using the equation *E* = *w_1_E_rep_* + *w_2_E_attr_* + *w_3_E_elec_* + *w_4_E_DARS_*_._ The PIPER algorithm has demonstrated strong performance in CAPRI assessments, calculating docking energies on a grid using the Fast Fourier Transform (FFT) and pairwise interaction potentials, which enables the rapid identification of near-native conformations [36]. The resulting structures were then clustered based on pairwise RMSD to optimize their representation as follows [4,66]. The top-ranked models were subsequently visualized using PyMol.

## 5. Conclusions

The expression of *Holsc*Activin and *Holsc*Inhibin was observed across all targeted tissues in the gene expression analysis and in situ hybridization. In the intestine and respiratory tree, these genes were identified in the intestinal and coelomic epithelium, connective tissues, and epithelial cells. Both genes were also present in spermatocytes within the testis and in oocytes of the ovary. Furthermore, their localization was specific to the connective tissue cells and fibers of the inner body wall. *Holsc*TGFBR exhibited a similar expression pattern, except for its absence in the respiratory tree. Phylogenetic and structural analyses revealed that *Holsc*TGFBR is evolutionarily conserved, sharing similarities with human TGF-β receptors. The presence of conserved sequence motifs and structural domains supports the functional significance of *Holsc*TGFBR in TGF-β signaling. Molecular docking studies demonstrated that *Holsc*Activin binds to *Holsc*TGFBR, potentially functioning as an agonist, whereas *Holsc*Inhibin likely inhibits receptor activation. The ternary complex interactions suggested a strong role for *Holsc*Inhibin and *Holsc*Activin in competitive TGF-β signaling, potentially contributing to tumor suppression and immune response enhancement. These findings illuminate the conserved and distinct mechanisms between *H. scabra* and human TGF-β signaling. Such insights have significant implications for health-related applications and the development of functional foods in the future.

## Figures and Tables

**Figure 1 ijms-26-06998-f001:**
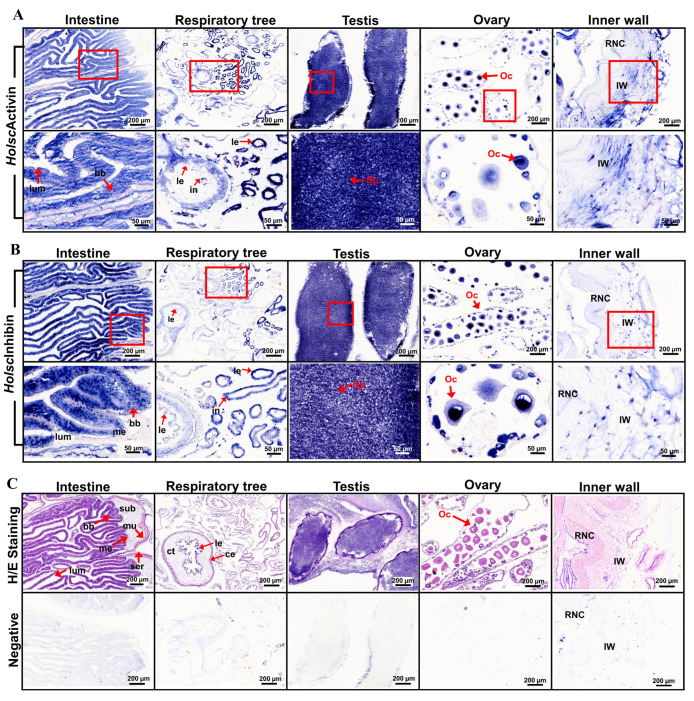
(**A**,**B**) In situ hybridization of *Holsc*Activin and *Holsc*Inhibin, respectively, in various sea cucumber organs. Positive cell localization is indicated by blue staining. (**C**) Upper: H/E stain is the section stained with hematoxylin and eosin. Below: Negative control of hybridization was performed with a DIG-labeled sense strand probe. The red box is represented on the right at higher magnification. Submucosa (sub), serosa (ser), muscle (mus), mucosa (muc), lumen (lum), brush borders (bb), oocyte (Oc), spermatocyte (Sp), coelomic epithelium (ce), muscular layer (ml), connective tissue (ct), infusoria (in), lining epithelium (le), mucosal epithelium (me), inner wall (IW), and radial nerve cord (RNC).

**Figure 2 ijms-26-06998-f002:**
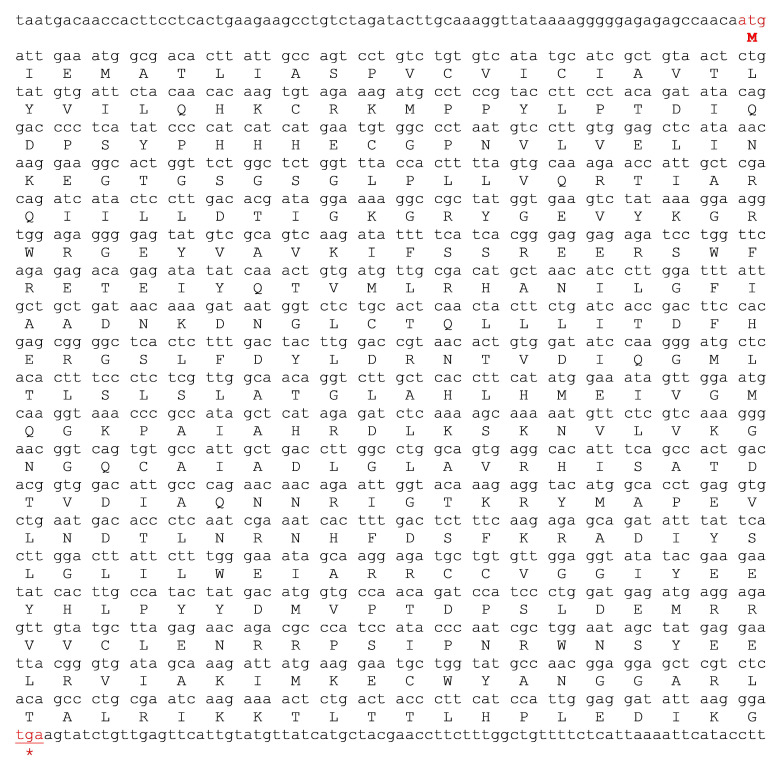
The nucleotide sequences and the deduced amino acid sequences of the *Holsc*TGFBR. The red letter indicates the first amino acid of the mature protein, and (*) represents the stop codon.

**Figure 3 ijms-26-06998-f003:**
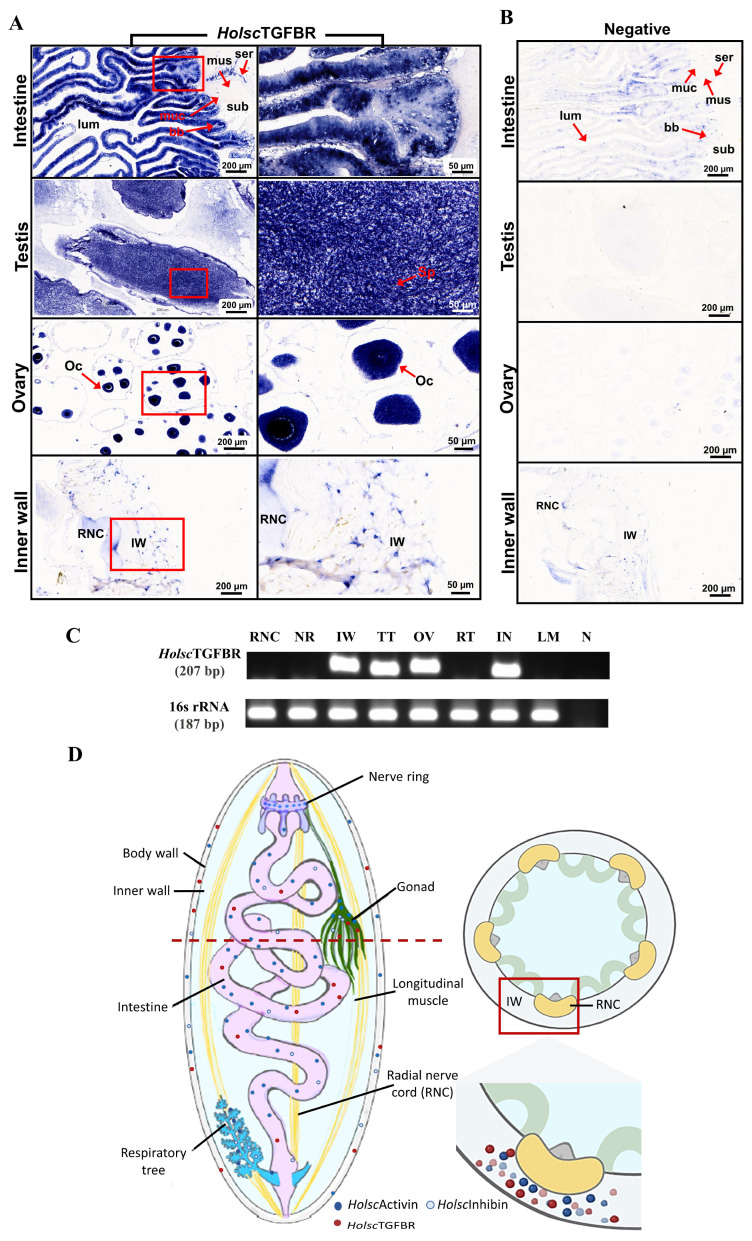
(**A**) In situ hybridization of *Holsc*TGFBR was specifically localized in various organs of the *H. scabra*. (**B**) Negative control of hybridization was performed with a DIG-labeled sense strand probe. (**C**) Tissue-specific expression of *Holsc*TGFBR using RT-PCR; the 16s rRNA expression was used as a housekeeping gene. RNC: radial nerve cord, NR: nerve ring, IW: inner wall, TT: testis, OV: ovary, RT: respiratory tree, IN: intestine, LM: longitudinal muscle, N: non-template control. (**D**) The illustration of *H. scabra* TGF-β proteins and receptor distribution in the inner wall tissue. Submucosa (sub), serosa (ser), muscle (mus), mucosa (muc), lumen (lum), brush borders (bb), oocyte (Oc), and spermatocyte (Sp). The red box shows a magnified view of the interactions between *Holsc*Activin and *Holsc*Inhibin with *Holsc*TGFBR.

**Figure 4 ijms-26-06998-f004:**
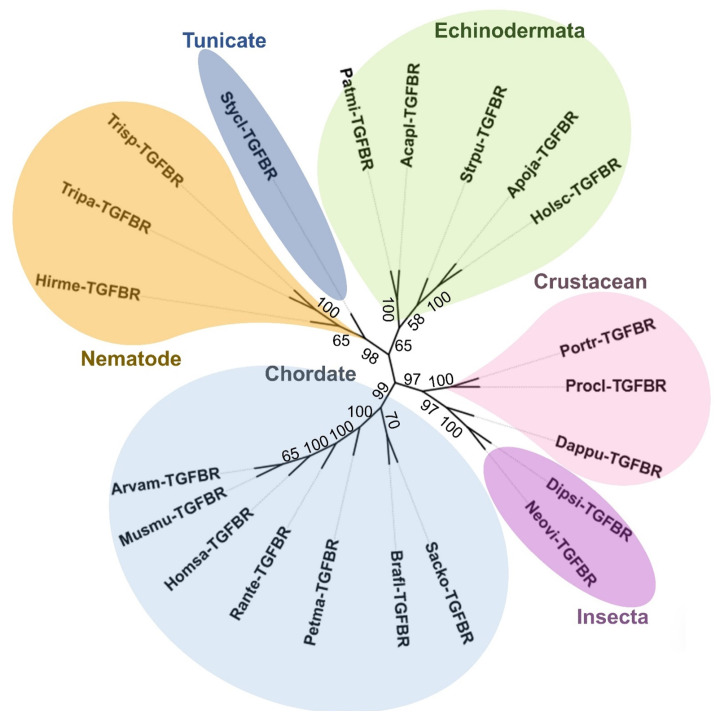
Phylogenetic analysis comparison of TGF-β receptor (TGFBR) and various species, including echinoderms, chordates, arthropods, crustaceans, nematodes, tunicates, and hemichordates *Holothuria scabra* (*Holsc*TGFBR), *Apostichopus japonicus* (*Apoja*TGFBR), *Acanthaster planci* (*Acapl*TGFBR), *Patiria miniata* (*Patmi*TGFBR), *Strongylocentrotus purpuratus* (*Strpu*TGFBR), *Styela clava* (*Stycl*TGFBR), *Trichinella* sp. (*Trisp*TGFBR), *Trichinella papuae* (*Tripa*TGFBR), *Hirudo medicinalis* (*Hirme*TGFBR), *Homo sapiens* (*Homsa*TGFBR), *Petromyzon marinus* (*Petma*TGFBR), *Mus musculus* (*Musmu*TGFBR), *Arvicola amphibius* (*Arvam*TGFBR), *Rana temporaria* (*Rante*TGFBR), *Branchiostoma floridae* (*Brafl*TGFBR), *Saccoglossus kowalevskii* (*Sacko*TGFBR), *Neodiprion virginiana* (*Neovi*TGFBR), *Diprion similis* (*Dipsi*TGFBR), *Daphnia pulicaria* (*Dappu*TGFBR), *Procambarus clarkii* (*Procl*TGFBR), and *Portunus trituberculatus* (*Portr*TGFBR).

**Figure 5 ijms-26-06998-f005:**
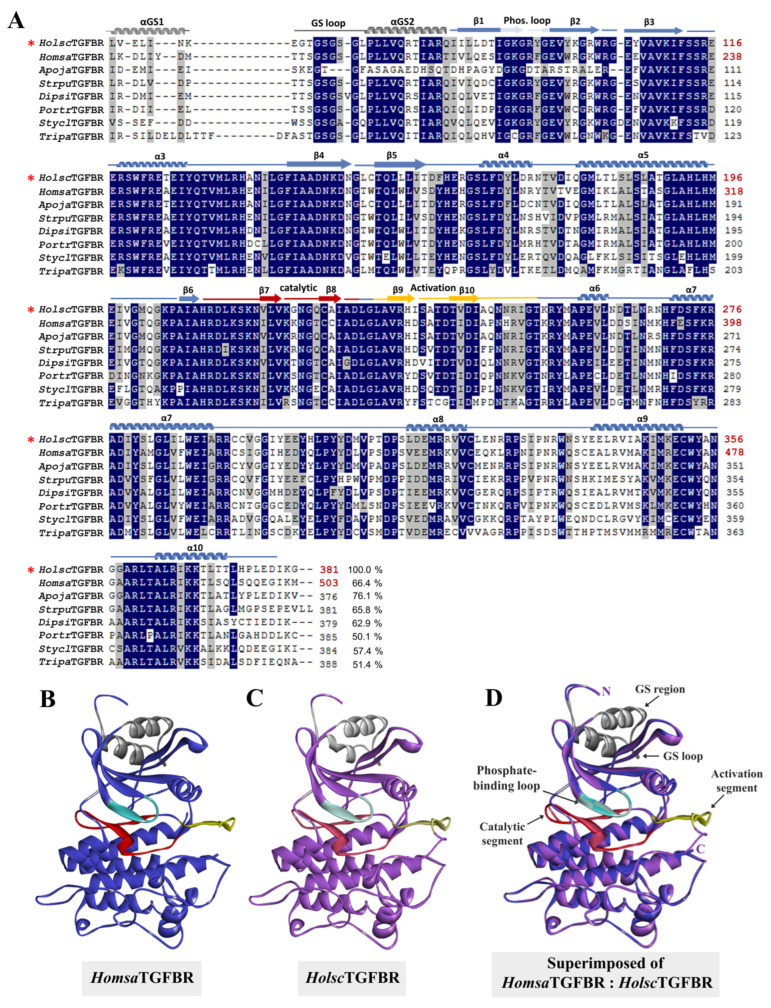
(**A**) Amino acid sequence alignment of *Holsc*TGFBR (red asterisk) and TGFBR from various species. The conserved and similar amino acid residues are labeled in dark blue and light blue, respectively. The numbers on the right indicate amino acid numbers. (**B**) *Homsa*TGFBR (template, PDB ID: 1b6c.3.B, blue ribbon) and (**C**) *Holsc*TGFBR (model, purple ribbon). (**D**) Superimposition of human Type I TGF-β Receptor (*Homsa*TGFBR) and *Holsc*TGFBR. The GS regions and loop, phosphate-binding, catalytic segment, and activation segments were colored as gray, cyan, red, and yellow, respectively.

**Figure 6 ijms-26-06998-f006:**
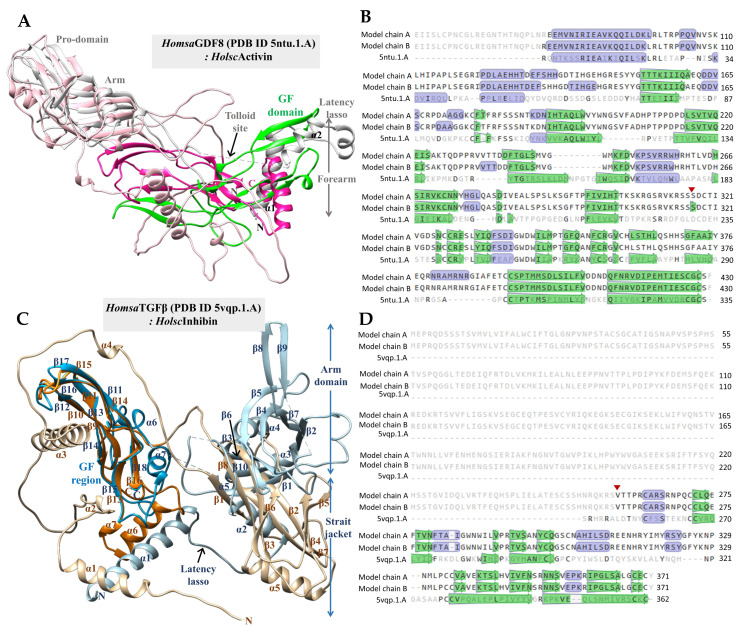
(**A**) Structural superimposition of monomeric *Homsa*GDF8 (human pro-myostatin precursor; 2.6 Å resolution; PDB ID: 5ntu.1.A; gray and green ribbons) and monomeric *Holsc*Activin (pale pink and deep pink ribbons). (**B**) Amino acid sequence alignment of *Homsa*GDF8 and *Holsc*Activin (model chains A and B). (**C**) Structural superimposition of *Homsa*TGFβ (pro-TGF-β1, PDB ID: 5vqp.1.A, pale blue and deep blue ribbons) and *Holsc*Inhibin (tan and orange ribbons). (**D**) Amino acid sequence alignment of *Homsa*TGFβ and *Holsc*Inhibin (model chains A and B). The red triangle indicates the start residues of mature proteins: G230–S335 for *Homsa*GDF8, S316–F430 for *Holsc*Activin, L253–S363 for *Homsa*TGFβ, and V258–Y371 for *Holsc*Inhibin.

**Figure 7 ijms-26-06998-f007:**
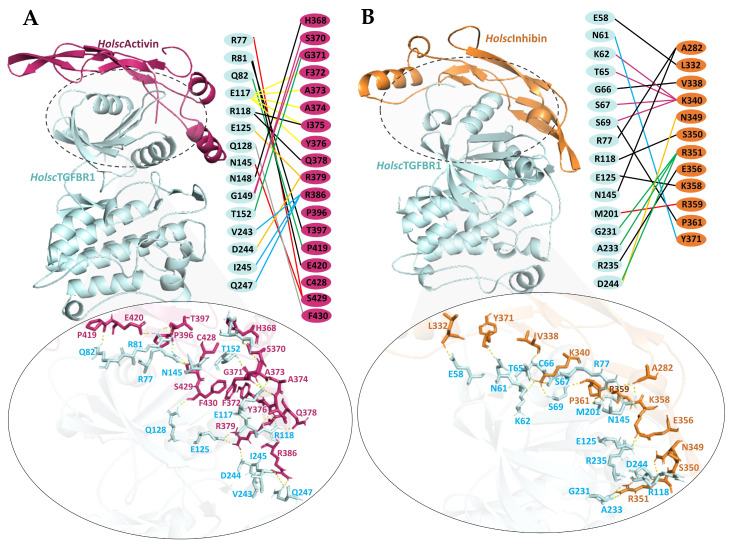
Molecular docking analysis of mature *H. scabra* TGF-β proteins and *Holsc*TGFBR complexes using *Homsa*TGFBR as the receptor template (PDB ID: 1b6c.3.B). (**A**) *Holsc*Activin–TGFBR (deep pink and light blue ribbons), (**B**) *Holsc*Inhibin–TGFBR (orange and light blue ribbons). The right panel coloring lines represent the TGF-β proteins and TGFBR interaction. Oval insert, yellow dotted lines represent hydrogen bonds.

**Figure 8 ijms-26-06998-f008:**
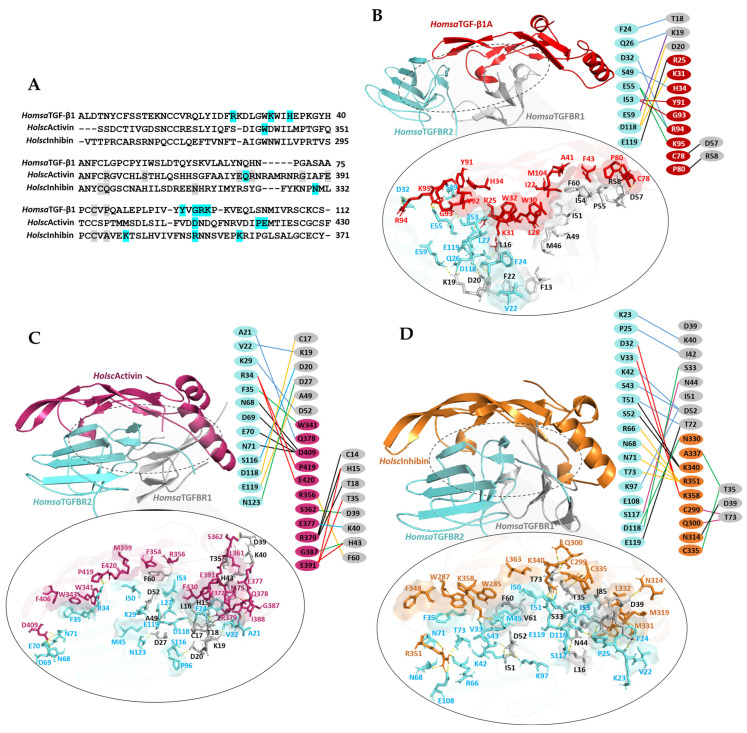
(**A**) Sequence analysis shows conserved residues in *Homsa*TGF-β1, *Holsc*Activin, and *Holsc*Inhibin. Cyan and gray highlights indicate hydrogen bonds with TGFBR2 and TGFBR1, respectively, based on docking results. (**B**) The ternary docking complex of *Homsa*TGF-β3 (red), *Homsa*TGFBR2 (blue), and *Homsa*TGFBR1 (gray) shows extensive interfacial interactions, obtained from crystal structure (PDB ID: 3KFD). (**C**,**D**) The ternary complex of *Holsc*Activin (deep pink) and *Holsc*Inhibin (orange) that were docked with *Homsa*TGFBR2 (blue) and *Homsa*TGFBR1 (gray). In the right panels of (**B**–**D**), the colored lines represent the hydrogen bonding interactions between the TGF-β proteins and the receptors. Oval insert, yellow dotted lines and the surface represent hydrogen bonds and hydrophobic interaction, respectively.

**Table 2 ijms-26-06998-t002:** Gene-specific primer sequences for molecular cloning and RT-PCR of *Holsc*TGFBR.

Gene	Primers	Primer Sequences (5′–3′)	Size (bp)	Purpose
*Holsc*TGFBR	HscTGFR-F6	TGGCGACACTTATTGCCAGT	101	Gene validation
	HscTGFR-R6	AGGAAGGTACGGAGGCATCT		
*Holsc*TGFBR	HscTGFR-F1	AGATGCCTCCGTACCTTCCT	580	Gene validation
	HscTGFR-R1	TGACCGTTCCCTTTGACGAG		
*Holsc*TGFBR	HscTGFR-F2	TCGTCAAAGGGAACGGTCAG	652	Gene validation
	HscTGFR-R2	TGGGTCAGCAAAACACCCTT		
*Holsc*TGFBR	HscTGFR-F4	AAGGGTGTTTTGCTGACCCA	280	Gene validation
	HscTGFR-R4	AGTGGCGTAAGTCCCAACTG		
*Holsc*TGFBR	HscTGFRRT-F	GGGGAGTATGTCGCAGTCAAGA	207	Gene expression
	HscTGFRRT-R	ACTGACCGTTCCCTTTGACGAG		
16s rRNA	16SF	GAAAGACGAGAAGACCCTGTCGAG	187	Gene expression
	16SR	CTTTTTCCGATTACCAGTTTCTGGTTC		

## Data Availability

The original contributions presented in this study are included in this article. Further inquiries can be directed to the corresponding author.

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
