# Peer review of "Characterization and Expression of TGF-β Proteins and Receptor in Sea Cucumber (Holothuria scabra): Insights into Potential Applications via Molecular Docking Predictions"

_ijms, 2025, doi:10.3390/ijms26146998_

Round 1

Reviewer 1 Report

Comments and Suggestions for Authors

The manuscript titled “Characterization and Expression of TGF-β Proteins and Receptor in Sea Cucumber (Holothuria scabra): Insights into Potential Applications via Molecular Docking Predictions” presents a compelling and novel investigation into the TGF-β signaling components in H. scabra. The work is well-conceived and contributes valuable insights into the molecular biology of a species of increasing interest due to its medicinal and nutritional relevance.

The combination of gene expression analysis, sequence conservation studies, and molecular docking provides a comprehensive approach to characterizing these proteins. The localization of HolscActivin, HolscInhibin, and HolscTGFBR across tissues is particularly informative, aligning well with their proposed physiological roles. The findings regarding structural and functional conservation with human homologs are also intriguing and suggest possible translational potential in biomedical or nutraceutical applications.

Overall, the manuscript is clearly written, scientifically sound, and appropriately structured. I found the study to be of high quality, with only minor points that require attention.

The discussion section contained several typographical errors in the orientation of parentheses (e.g., lines 303, 306), including those in the reference citations. Please review and correct them to maintain clarity and consistency.

Additionally, while the molecular docking analysis using ClusPro 2.0 adds a valuable dimension to the study, the methods section would benefit from more detail. Specifically, it would be helpful to know whether the authors used default ClusPro settings or made any modifications to the docking parameters, particularly regarding selection of binding poses, scoring metrics, or treatment of flexible regions. Clarifying these points would enhance the reproducibility and interpretability of the docking results.

In summary, I recommend acceptance of this manuscript pending minor revisions. The study presents important findings with relevance to both basic science and potential applied research, and it is a welcome contribution to the field.

Author Response

Comment: The discussion section contained several typographical errors in the orientation of parentheses (e.g., lines 303, 306), including those in the reference citations. Please review and correct them to maintain clarity and consistency.

Response: We have checked and revised these errors already.

Comment: Additionally, while the molecular docking analysis using ClusPro 2.0 adds a valuable dimension to the study, the methods section would benefit from more detail. Specifically, it would be helpful to know whether the authors used default ClusPro settings or made any modifications to the docking parameters, particularly regarding selection of binding poses, scoring metrics, or treatment of flexible regions. Clarifying these points would enhance the reproducibility and interpretability of the docking results.

Response: Thank you for your suggestions. We have added more method details for this concern.

Line: 555-571 (in main text)

Complex structures of HolscActivin and HolscInhibin with the HolscTGFBR were docked using the cytoplasmic domain of the type I TGF-β receptor (PDB ID: 1b6c) as a complex structural template. For ternary complex docking, HomsaTGFβ was initially docked with HomsaTGFBR2. The complexes with the most favorable (lowest) energy scores were selected, and HomsaTGFBR1 was subsequently recruited to form the complete ternary complex. The crystal structure of the ternary complex (PDB ID: 3KFD) was used as the experimental template. This protocol was similarly applied using HolscActivin and HolscIn-hibin in place of HomsaTGFβ. All molecular docking complexes were performed on ClusPro 2.0, a widely recognized server for protein-protein docking (https://cluspro.org/help.php, accessed on 6 July 2025). The interaction energy of the protein complexes is based on the PIPER protein interaction energy, using the equation E = w1Erep + w2Eattr + w3Eelec + w4EDARS. The PIPER algorithm has demonstrated strong performance in CAPRI assessments, calculating docking energies on a grid using the Fast Fourier Transform (FFT) and pairwise interaction potentials, which enables the rapid identification of near-native conformations [36]. The resulting structures were then clustered based on pairwise RMSD to optimize their representation as follows [4,66]. The top-ranked models were subsequently visualized using PyMol.

Reviewer 2 Report

Comments and Suggestions for Authors

This manuscript reports the original investigation of detecting gene expression of TGF-β proteins (activin and inhibin) and type I TGF-β receptor in various tissues and organs of the sea cucumber (H. scabra), structural prediction of H. scabra TGF-β proteins (activin and inhibin) and type I TGF-β receptor by using homology modeling, as well as investigation of protein-protein interactions between H. scabra TGF-β proteins (activin and inhibin) and the cytoplasmic domain of human type I TGF-β receptor by using protein-protein molecular docking.

According to literatures, activin and inhibin do not directly bind to the cytoplasmic domain of type I TGF-β receptor. Otherwise, activin or inhibin binds to the extracellular domain of type II TGF-β receptor, and the complex binds and recruits type I TGF-β receptor. Thus, the rationale of homologous structure prediction model of type I TGF-β receptor using the structure of cytoplasmic domain of human type I TGF-β receptor and subsequent molecular dockings will be compromised.   

Comments:

#1: To investigate interactions between TGF-β proteins and TGF-β receptors, protein-protein molecular dockings should be performed between TGF-β proteins and type II TGF-β receptor. For the type II TGF-β receptor, the extracellular domain must be included in the structure.

#2: Validation is required to evaluate protein-protein molecular dockings, such as including positive control proteins and/or negative control proteins (if applicable).

#3: Since 26% to 29% of sequence identities fall in the twilight zone, the homology models should be used with caution. It is highly recommended to use Alphafold prediction models for both structural comparison and protein-protein molecular dockings.

Author Response

Comment #1: To investigate interactions between TGF-β proteins and TGF-β receptors, protein-protein molecular dockings should be performed between TGF-β proteins and type II TGF-β receptor. For the type II TGF-β receptor, the extracellular domain must be included in the structure.

Response: Thank you for your valuable suggestions. These comments significantly improve our work. Therefore, we re-examined the docking of H. scabra TGF-β proteins, as well as human TGF-β, interacting with human TGFBR2, followed by recruitment of TGFBR1 to mimic endogenous TGF-β signal transduction. In this stage, we revised the methods, results, and discussion accordingly, as highlighted in yellow in the main text.

Comment #2: Validation is required to evaluate protein-protein molecular dockings, such as including positive control proteins and/or negative control proteins (if applicable).

Response: For validation of the docking method, we used the crystal structure of the ternary complex (PDB ID: 3KFD) as the experimental template or positive control. The ternary complex is composed of HomsaTGF-β1, HomsaTGFBR2, and HomsaTGFBR1. The complete complex exhibited both hydrogen bonding and hydrophobic interactions at regions and specific residues consistent with those in the experimental crystal structure (Radaev et al., 2010). This approach can therefore be applied to HolscActivin and HolscInhibin for cross-species docking analyses, providing further insights into the potential function of H. scabra proteins.

Comment #3: Since 26% to 29% of sequence identities fall in the twilight zone, the homology models should be used with caution. It is highly recommended to use Alphafold prediction models for both structural comparison and protein-protein molecular dockings.

Response: We appreciate your comment. We regenerated HolscActivin and HolscInhibin approached on AlphaFold2 (https://colab.research.google.com/github /sokrypton/ColabFold /blob/main/AlphaFold2. ipynb). Moreover, after predicting the 3D structures, HolscActivin and HolscInhibin were analyzed and structurally aligned using the crystal structures of the closest human homologs as templates, including HomsaGDF8 (human pro-myostatin precursor; 2.6 Å resolution, PDB ID: 5ntu.1.A) for HolscActivin, and HomsaTGFβ (human pro-TGF-β1, PDB ID: 5vqp.1.A) for HolscInhibin. Additionally, the mature forms of HolscActivin and HolscInhibin were predicted based on their structural templates and subsequently used as mimic TGF ligands in the docking complexes.

All details are provided in the main text and highlighted in yellow.

Reference:

Radaev S, Zou Z, Huang T, Lafer EM, Hinck AP, Sun PD. Ternary complex of transforming growth factor-beta1 reveals isoform-specific ligand recognition and receptor recruitment in the superfamily. J Biol Chem. 2010 May 7;285(19):14806-14. doi: 10.1074/jbc.M109.079921.

Reviewer 3 Report

Comments and Suggestions for Authors

The paper by Nonkhwao et al. investigates the localisation, sequence conservation, structural alignment, and potential functions of Holothuria scabra TGF-β proteins (activin and inhibin) and their receptor across multiple organs. In addition, the paper makes a comparison between H. scabra TGF-β receptors and those of other species, especially Homo sapiens, and assesses cross-species binding via molecular docking.

The research project was meticulously designed, with the authors conducting a comprehensive evaluation of the in situ hybridisation technique to localise activin, inhibin, and TGF-β receptor mRNAs in various anatomical regions, including the intestine, respiratory tree, ovary, testis, and inner body wall. The project utilised RT-PCR to undertake tissue-specific expression profiling of the H. scabra TGF-β receptor, facilitating an in-depth analysis of evolutionary conservation through phylogenetic and structural analyses. Additionally, a molecular docking study was conducted, exploring the interaction between H. scabra ligands and both HolscTGFBR and human TGF-β receptor type I, offering valuable insights into the molecular mechanisms underlying these processes.

The authors of the study established that the TGF-β ligands and receptors of H. scabra are structurally and functionally conserved with their vertebrate counterparts. This finding highlights their potential to regulate growth, differentiation and immune responses. The sea cucumber proteins thus represent promising candidates for functional food ingredients or biomedical applications.

As the authors have asserted, the progression of therapeutic development is contingent upon the conduction of additional studies that address the following: safety, efficacy, human receptor interactions, and targeted delivery.

The principal disadvantage of the paper is the language employed, which in certain instances renders it arduous to comprehend. Moreover, there are instances of sentences that lack verbs, thus resulting in their failure to convey any substantial meaning.

It is evident that the paper needs minor revisions with respect to its formal presentation.

Author Response

Comment: The principal disadvantage of the paper is the language employed, which in certain instances renders it arduous to comprehend. Moreover, there are instances of sentences that lack verbs, thus resulting in their failure to convey any substantial meaning.

It is evident that the paper needs minor revisions with respect to its formal presentation.

Response: We appreciate your kind suggestions. We have had the manuscript professionally edited for English language by American Manuscript Editors service. The revised words and sentences are shown in red text throughout the main document.

Round 2

Reviewer 2 Report

Comments and Suggestions for Authors

This revised manuscript has addressed comments put forth in previous review. The manuscript is accepted for publication.